# Impact of *CDKN2A/B* Homozygous Deletion on the Prognosis and Biology of IDH-Mutant Glioma

**DOI:** 10.3390/biomedicines10020246

**Published:** 2022-01-24

**Authors:** L. Eric Huang

**Affiliations:** 1Department of Neurosurgery, Clinical Neurosciences Center, University of Utah, Salt Lake City, UT 84132, USA; eric.huang@hsc.utah.edu; 2Department of Oncological Sciences, Huntsman Cancer Institute, University of Utah, Salt Lake City, UT 84112, USA

**Keywords:** *CDKN2A/B*, cell cycle, glioma, IDH mutation, immunotherapy, stem-like cell, *TP53*, tumor-suppressor gene, WHO classification

## Abstract

Although hotspot mutations in isocitrate dehydrogenase (IDH) genes are associated with favorable clinical outcomes in glioma, *CDKN2A/B* homozygous deletion has been identified as an independent predicator of poor prognosis. Accordingly, the 2021 edition of the World Health Organization (WHO) classification of tumors of the central nervous system (CNS) has adopted this molecular feature by upgrading IDH-mutant astrocytoma to CNS WHO grade IV, even in the absence of glioblastoma-specific histological features—necrosis and microvascular proliferation. This new entity of IDH-mutant astrocytoma not only signifies an exception to the generally favorable outcome of IDH-mutant glioma, but also brings into question whether, and, if so, how, *CDKN2A/B* homozygous deletion overrides the anti-tumor activity of IDH mutation by promoting the proliferation of stem/neural progenitor-like cells. Understanding the mechanism by which IDH mutation requires intact tumor-suppressor genes for conferring favorable outcome may improve therapeutics.

## 1. Adoption of *CDKN2A/B* Homozygous Deletion in the Latest WHO Classification

Building on the 2016 edition of WHO classification of CNS tumors and the recommendation of the cIMPACT-NOW (Consortium to Inform Molecular and Practical Approaches to CNS Tumor Taxonomy—Not Official WHO), the 2021 edition of WHO classification further advances the role of molecular diagnostics in CNS tumor classification [1,2,3]. The molecular parameter of IDH status—either presence or absence of recurrent mutations in the *IDH1* and *IDH2* genes—was first adopted in the 2016 edition of WHO classification [1]. This molecular feature defined a major characteristic of diffuse astrocytic and oligodendroglial tumors: IDH-mutant gliomas are associated with distinct biology and favorable clinical outcomes, whereas IDH-wildtype gliomas share genomic aberrations and clinical behavior with glioblastomas [4]. The inclusion of 1p/19q codeletion and histology features further classified IDH-mutant gliomas into 1p/19q-codeleted oligodendroglioma or anaplastic oligodendroglioma, diffuse astrocytoma or anaplastic astrocytoma, and glioblastoma [1] (Table 1).

Although the 2016 edition of WHO classification successfully distinguished the three histologic subtypes of IDH-mutant gliomas in overall survival [4], enormous variability remained within the IDH-mutant astrocytoma group. Subsequently, the cIMPACT-NOW recommended upgrading those harboring homozygous deletion of *CDKN2A* (cyclin-dependent kinase inhibitor 2A) and *CDKN2B* (abbreviated as *CDKN2A/B*), either alone or in combination with microvascular proliferation or necrosis, to a newly described entity: IDH-mutant astrocytoma, WHO grade IV (to be distinguished from glioblastoma, WHO grade IV) [2]. This recommendation was based on multiple clinical studies indicating that *CDKN2A/B* homozygous deletion is a strong adverse prognostic factor, as this genetic alteration renders IDH-mutant astrocytoma ~50% (61 v. > 120 months) to 68% (52 v. 165 months) shorter in median overall survival compared to those without *CDKN2A/B* homozygous deletion and virtually indistinguishable from IDH-mutant glioblastoma [5,6,7]. Furthermore, multivariate analyses confirmed that *CDKN2A/B* homozygous deletion is a strong predictor of shorter progression-free survival and overall survival [7].

Accordingly, the latest 2021 edition of WHO classification has officially adopted *CDKN2A/B* homozygous deletion as the sole molecular feature in IDH-mutant astrocytoma, with the recommendation of an integrated and layered diagnosis: astrocytoma, IDH-mutant, CNS WHO grade IV, *CDKN2A/B* homozygous deletion [3,8] (Table 1). Of note, this new entity is distinct from IDH-wildtype astrocytomas harboring one or more of the molecular features *TERT* promoter mutation, *EGFR* amplification, gain of entire chromosome 7, or loss of entire chromosome 10, which are diagnosed collectively as Glioblastoma, IDH-wildtype, CNS WHO grade IV with the molecular information (Table 1). Of note, NEC (Not Elsewhere Classified) or NOS (Not Otherwise Specified) is appended to diagnoses either nonconforming to the WHO classification or lacking molecular specification.

## 2. Impact of *CDKN2A/B* Homozygous Deletion on the Biology of IDH Mutation

*CDKN2A* is localized in chromosome 9p21.3 and encodes two tumor-suppressor proteins that regulate the activities of p53 and pRB (encoded by *TP53* and *RB1* genes, respectively) in tumor suppression; ARF (alternate reading frame; aka p14^ARF^ in human) triggers p53-mediated cell-cycle arrest or apoptosis by inactivating MDM2 (mouse double minute 2), an E3 ubiquitin-protein ligase targeting p53 for destabilization, whereas INK4a (aka p16^INK4a^) promotes pRB-mediated cell-cycle checkpoints by inhibiting CDK4 (cyclin d-dependent kinase 4) that phosphorylates and inactivates pRB [9]. At the same locus, *CDKN2B* encodes INK4B (aka p15^INK4B^), another inhibitor of CDKs that controls cell proliferation by inactivating CDK4/CDK6. Furthermore, p53 and pRB form a feedback control loop, where the p53 transcriptional target *CDKN1A* (encoding p21^Cip1/Waf1^) reverts pRB to a hypophosphorylated, growth-inhibitory state.

Although the biological function of IDH mutation in glioma remains debatable, i.e., oncogenic or tumor-suppressive [10], the importance of *CDKN2A/B* homozygous deletion in patient survival may help resolve the controversy. Owing to the technical difficulties in maintaining bona fide IDH-mutant cells in culture [11], numerous studies have resorted to cell models harboring either *CDKN2A/B* homozygous deletion or inactivated *TP53* and *RB1* tumor-suppressor genes, including the “normal human astrocytes” (NHA), which are transduced with the human papillomavirus 16 E6/E7 oncoproteins to block p53 and pRB signaling [12], and the glioblastoma U-87MG cell line, which harbors *CDKN2A/B* homozygous deletion and *PTEN* loss [13]. By relying on these models, IDH mutation has been shown to initiate oncogenic transformation and epigenetic reprogramming through DNA and histone hypermethylation, and to reduce tumor-free survival [14,15,16,17,18]. Moreover, these studies may inadvertently provide a mechanistic endorsement of upgrading to CNS WHO grade 4 of IDH-mutant astrocytoma harboring *CDKN2A/B* homozygous deletion.

It has been speculated, however, that the biological function of IDH mutation may be skewed by the inactivation of both *TP53* and *RB1* genes [10], as multiple studies have demonstrated that, in the absence of *CDKN2A/B* homozygous deletion, IDH mutation inhibited glioma genesis and extended survival in comparison with wild-type IDH [19,20,21]. In particular, *Cdkn2a^+/+^* mice with *IDH1*-mutant glioma had significantly longer median survival; however, they completely lost the survival advantage—and indeed faced a greater reduction in median survival—upon genetic deletion (Figure 1). Therefore, these findings not only corroborate the detrimental effect of *CDKN2A* homozygous deletion on patient survival of IDH-mutant astrocytoma, but also suggest its negative impact on the tumor-suppressive activity of IDH mutation, either directly or indirectly.

The tumor-suppressive activity of IDH mutation is indicated by the finding that D-2-hydroxyglutarate—the distinct metabolite produced from IDH mutations—exerts an anti-tumor activity by attenuating aerobic glycolysis in leukemia cells [22]. In keeping with this, IDH mutations specifically producing higher levels of D-2-hydroxyglutarate are associated with better survival in astrocytoma patients [23,24]. Furthermore, various animal models with *IDH1* or *IDH2* mutation recapitulated neurodegeneration [20,25,26,27], but not glioma genesis despite the epigenetic and transcriptomic resemblance [27]. Moreover, germline *IDH2* mutations phenocopied cardiomyopathy and muscular dystrophy [26], as seen in D-2-hydroxyglutaric aciduria patients harboring autosomal dominant *IDH2* mutations [28]. Collectively, all these genetic models phenocopy human’s pathological lesions except tumors. In fact, IDH mutations have also been found in healthy human tissues: *IDH1* mutations in the glial cells of younger individuals and *IDH2* mutations in the heart and skeletal muscle [29,30]. Taken together, the integrity of *CDKN2A/B* distinguishes the biological outcomes of IDH mutation during the course of glioma progression, in accordance with its impact on patient survival.

## 3. Dependence of Tumor-Suppressor Genes on the Biology of IDH Mutation

Although why IDH mutation requires intact *CDKN2A/B* for conferring favorable clinical outcome remains to be investigated, its anti-tumor activity seemingly depends on the integrity of tumor-suppressor genes, as depicted in Figure 2; the activity is extinct upon *CDKN2A/B* homozygous deletion, weakened in the presence of *TP53* alteration or 1p/19q codeletion, and most potent when the tumor-suppressor genes remain intact. In other words, *CDKN2A/B* homozygous deletion represents a tipping point that tolls the knell for the favorable outcome of IDH-mutant glioma [5,6,7]. In the absence of *CDKN2A/B* homozygous deletion, the widespread *TP53* mutation in IDH-mutant astrocytoma is associated with shortened overall survival when compared with IDH-mutant astrocytoma of *TP53*-wildtype [24], a finding conforming to the importance of tumor-suppressor genes in cancer biology. In the *Trp53*, *Cdkn2a*-intact background, however, *IDH1* mutation exerts potent tumor suppression by abrogating oncogene-induced glioma genesis [31]. Therefore, loss of tumor-suppressor genes, notably *CDKN2A/B*, nullifies the tumor-suppressive activity of IDH mutation, resulting in progression to aggressive IDH-wildtype-like glioma (Figure 2).

Although DNA methylome profiling has been recognized as a powerful approach to CNS tumor classification, it has yet to be integrated into the WHO classification [3,32]. Nevertheless, this approach has identified a subset of IDH-mutant glioma as G-CIMP (glioma-CpG island methylator phenotype) low [33,34,35]. This G-CIMP low group is characterized by poor clinical outcome and genetic abnormalities in *CDKN2A* and *CDKN4,* accompanied by cell-cycle gene activation. Therefore, increased cell proliferation, owing to the genetic and/or epigenetic alterations, overrides the anti-proliferative activity of IDH mutation to drive glioma progression (Figure 2). This notion is supported by the findings from single-cell analyses that IDH-mutant glioma exhibits restricted cell proliferation with differentiation outpacing dedifferentiation in a developmental hierarchy of malignant cells [36,37,38,39]. The developmental hierarchy comprises three subpopulations: the proliferative, undifferentiated stem/neural progenitor-like cells, the non-proliferative, differentiated astrocyte-like cells, and the oligodendrocyte-like cells [36,37,38]. In IDH-mutant glioma, proliferation is primarily restricted to the rare stem/neural progenitor-like subpopulation, in contrast to the high percentages of proliferative cells in IDH-wildtype glioblastoma, which manifests cellular state heterogeneity and greater plasticity [38,40].

## 4. Targeting IDH-Mutant Glioma

Multiple trials of various mutant IDH inhibitors for glioma are currently underway [41,42] following the promising results of a phase 1 trial [43]. Although these drugs are effective in reducing D-2-hydroxyglutarate levels and inducing cell differentiation, and some are brain penetrant, the clinical outcomes still remain to be seen for the following reasons [41,42,44,45,46]. First, both IDH mutation and D-2-hydroxyglutarate are seemingly nonessential in glioma progression. Second, mutant IDH inhibitors may desensitize glioma cells due to increased NADPH production. Lastly, some of these drugs may have severe adverse events.

A recent study on pediatric high-grade glioma harboring histone H3.3 (*H3-3A*) G34R/V mutations has set an important precedent for uncovering bona fide oncogenic signaling to be potentially targetable [47]. Like IDH-mutant glioma, G34R/V glioma was thought to be driven by epigenomic reprogramming [48], as G34R/V promotes repressive trimethylation of histone 3 lysine 27, also seen in IDH-mutant glioma cells, to block neuronal differentiation [47]. It has been discovered, however, that the oncogenic *PDGFRA* gene drives tumorigenesis by hijacking lineage-specific regulatory elements in the stalled interneuron progenitors. Hence, G34R/V gliomas, in fact, arise from mis-regulation of interneuron differentiation that enables opportunistic activation of potently oncogenic PDGF signaling, which is potentially targetable, whereas G34R/V mutation appears dispensable for tumor maintenance. Although epigenomic reprogramming in *IDH1*-mutant glioma also results in aberrant *PDGFRA* expression through the dysfunction of methylation-sensitive insulator [49], whether or not this is the mechanism of IDH-mutant glioma genesis remains to be investigated.

Alternative strategies for targeting IDH-mutant glioma are well underway [11,50]. In particular, by targeting the clonal neoepitope, the mutant IDH1-specific peptide vaccine (IDH1-vac) has yielded, thus far, the best safety and efficacy as a single agent in newly diagnosed glioma in terms of overall response rate, progression-free survival, and overall survival compared to other trials of mutant IDH inhibitors for various types of cancer [42,51,52]. This mutation-specific vaccine elicited robust peripheral T cell responses and intratumoral inflammatory reactions, irrespective of any known tumor-intrinsic molecular markers, including *CDKN2A/B* homozygous deletion and methylation status. Of note, although mutation-based neoantigens are an attractive model for therapeutic vaccines, clonal loss of the IDH-mutant allele has also been noted in recurrent glioma [53], thereby a potential mechanism of resistance.

In sum, the prognostic importance of *CDKN2A/B* homozygous deletion in IDH-mutant glioma begs the question of how this genetic alteration impacts the biology of IDH mutation in relation to patient survival, and, importantly, how to develop a coherent strategy for the treatment of IDH-mutant glioma.

## Figures and Tables

**Figure 1 biomedicines-10-00246-f001:**
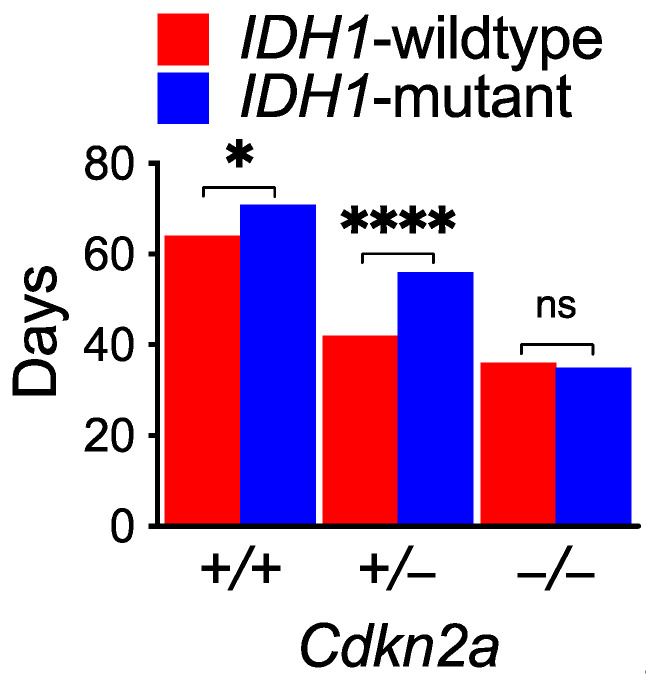
*Cdkn2a* homozygous deletion in mice abrogates the survival benefit of *IDH1*-mutant glioma. Comparison of median survival between mice of *IDH1*-wildtype glioma and *IDH1*-mutant glioma in different *Cdkn2a* backgrounds, using previously published data [19]. * *p* < 0.05; **** *p* < 0.001; ns, not significant.

**Figure 2 biomedicines-10-00246-f002:**
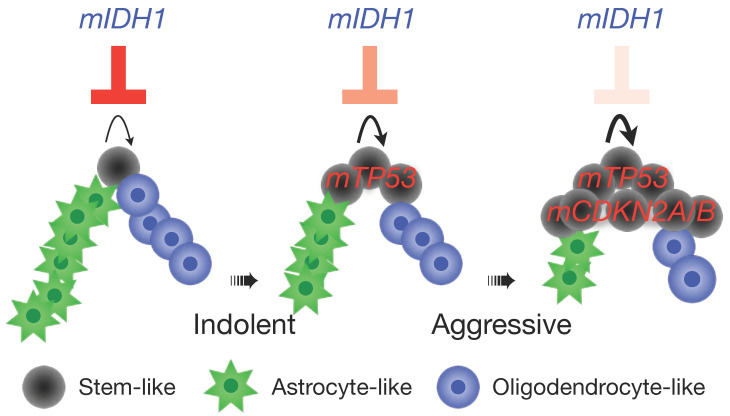
*IDH1* mutation-mediated inhibition of cell proliferation depends on intact tumor-suppressor genes. IDH-mutant glioma cells are depicted in a developmental hierarchy model where stem-like cells are proliferative. The anti-proliferative activity of *IDH1* mutation (*mIDH1*) is potent in the presence of intact tumor-suppressor genes, weakened by *TP53* alteration (*mTP53*), and lost upon *CDKN2A/B* homozygous deletion (*mCDKN2A/B*).

**Table 1 biomedicines-10-00246-t001:** Comparison between 2021 and 2016 WHO classifications of glioma.

Histology	WHO 2016	Grade	WHO 2021	Grade
Oligodendroglioma	Oligodendroglioma, **IDH**-mutant and **1p/19q**-codeleted	WHO grade II	Oligodendroglioma, **IDH**-mutant and **1p/19q**-codeleted	CNS WHO grade 2
Anaplastic oligodendroglioma	Anaplastic oligodendroglioma, **IDH**-mutant and **1p/19q**-codeleted	WHO grade III	Oligodendroglioma, **IDH**-mutant and **1p/19q**-codeleted	CNS WHO grade 3
Diffuse astrocytoma	Diffuse astrocytoma, IDH-wildtype or **IDH**-mutant	WHO grade II	Astrocytoma, **IDH**-mutant	CNS WHO grade 2
Anaplastic astrocytoma	Anaplastic astrocytoma, IDH-wildtype or **IDH**-mutant	WHO grade III	Astrocytoma, **IDH**-mutant	CNS WHO grade 3
Glioblastoma	Glioblastoma, IDH-wildtype or **IDH**-mutant	WHO grade IV	Glioblastoma, IDH-wildtype	CNS WHO grade 4
Astrocytoma			Astrocytoma, **IDH**-mutant and *CDKN2A/B* homozygous deletion	CNS WHO grade 4
Astrocytoma			Glioblastoma, IDH-wildtype & *TERT* **promoter** mutation, *EGFR* amplification, or gain/loss of **chromosome 7/10**	CNS WHO grade 4

Genetic alterations essential to the WHO classification are indicated in bold.

## Data Availability

Not applicable.

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
