# Peer review of "Impact of CDKN2A/B Homozygous Deletion on the Prognosis and Biology of IDH-Mutant Glioma"

_biomedicines, 2022, doi:10.3390/biomedicines10020246_

Round 1

Reviewer 1 Report

Mutations in the isocitrate dehydrogenase (IDH) genes and CDKN2A/B homozygous deletion have opposite prognoses on glioma. The manuscript attempted to describe how CDKN2A/B impacts the biology of IDH mutation in astrocytoma.

Figure 1 legend stated “Cdkn2a homozygous deletion in mice abrogates survival benefit of IDH1-mutant glioma.” However, it is equally possible that the two mutations are two mutually non-interactive mutations. It happens that Cdkn2a homozygous deletion is a more dominant, more detrimental mutation that overrides the beneficial effect of IDH mutation. There may be no realistic interactions between IDH1 and CDKN2A, at least the manuscript did not mention it. That is also why Figure 2 appears to be hollow without realistic information.

Although the statement “The anti-tumor activity of IDH1 mutation (mIDH1) is potent in the presence of intact tumor-suppressor genes, weakened by TP53 alteration (mTP53), and lost upon CDKN2A/B homozygous deletion (mCDKN2A/B)” is correct, the description falls short of extending to the reason why. The reason or the mechanism would appear to be the purpose of this manuscript.

The manuscript failed to convey how genetic alteration of CDKN2A/B homozygous deletion impacts the biology of IDH mutation, resulting in a poorer prognosis.

The abstract and the manuscript can include more of the IDH mechanism and its effect on gliomas.

Author Response

I greatly appreciate the helpful criticisms and suggestions from the reviewers. By including the recent studies on methylome profiling and single-cell analysis, the revision has provided a mechanistic model depicting the negative impact of CDKN2A deletion on IDH mutation-mediated inhibition of stem-like cell proliferation, thereby driving glioma progression.

Reviewer 1

“Mutations in the isocitrate dehydrogenase (IDH) genes and CDKN2A/B homozygous deletion have opposite prognoses on glioma. The manuscript attempted to describe how CDKN2A/B impacts the biology of IDH mutation in astrocytoma.”

Response: Despite the clinical discovery that CDKN2A/B homozygous deletion is a key molecular feature that abrogates the survival advantage of IDH-mutant astrocytoma, the biological studies have yet to fully appreciate the importance of this molecular change in the biology of IDH mutation.

“Figure 1 legend stated “Cdkn2a homozygous deletion in mice abrogates survival benefit of IDH1-mutant glioma.” However, it is equally possible that the two mutations are two mutually non-interactive mutations. It happens that Cdkn2a homozygous deletion is a more dominant, more detrimental mutation that overrides the beneficial effect of IDH mutation. There may be no realistic interactions between IDH1 and CDKN2A, at least the manuscript did not mention it. That is also why Figure 2 appears to be hollow without realistic information.”

Response: I agree with the interpretation that the effect of Cdkn2a homozygous deletion in IDH1-mutant glioma is more dominant, thereby overriding the beneficial effect of IDH1 mutation. Given the unknown relationship between these two mutations, I speculate Cdkn2a homozygous deletion either directly or indirectly impacts the tumor-suppressive activity of IDH mutation (see line 109). Moreover, Figure 2 has been revised to model the dependence of IDH mutation-mediated inhibition of stem-like cell proliferation on intact tumor-suppressor genes.

“Although the statement “The anti-tumor activity of IDH1 mutation (mIDH1) is potent in the presence of intact tumor-suppressor genes, weakened by TP53 alteration (mTP53), and lost upon CDKN2A/B homozygous deletion (mCDKN2A/B)” is correct, the description falls short of extending to the reason why. The reason or the mechanism would appear to be the purpose of this manuscript.”

Response: A new paragraph (line 157 to line 175) has been added to review recent publications on G-CIMP-low glioma, CDKN2A deletion, and a developmental hierarchy from single-cell analyses of IDH-mutant glioma. A mechanistic model where CDKN2A deletion abrogates IDH mutation-mediated inhibition of stem-like cell proliferation is presented in revised Figure 2.

“The manuscript failed to convey how genetic alteration of CDKN2A/B homozygous deletion impacts the biology of IDH mutation, resulting in a poorer prognosis.”

Response: Please see above.

“The abstract and the manuscript can include more of the IDH mechanism and its effect on gliomas.”

Response: Both Abstract and main text have been revised accordingly.

Reviewer 2 Report

In his revised manuscript „ The Favorable Outcome of IDH-Mutant Glioma: CDKN2A/B Changes the Rule” Huang reports on elemental renewals in the revised WHO classification of tumors of the central nervous system or more specifically the upgrade of IDH-mutant astrocytoma with concurrent homozygous CDKN2A/B deletion (HD) to CNS WHO grade IV. Moreover, the author carved out further insights regarding the pathophysiological biology of how CDKN2A/B HD lead to unfavorable outcomes in this entity. I really enjoyed reviewing this manuscript which is clearly written, well-structured and of clinical relevance. In order to ensure the validity of this review article, there is one minor point that need to be addressed prior to publication. From my perspective this manuscript provides an excellent overview of fundamental modifications concerning the latest WHO-classification of gliomas.

Minor:

  • There exist several comprehensive methylome-analyses of IDH-mt gliomas in literature describing methylation subgroups (C-GIMP high vs. C-GIMP low; PMID 31667475 and/or PMID 26824661). In order to describe underlying molecular mechanisms of the diverging clinical behavior in IDH-mutant gliomas and the dependence on CDKN2A/B HD the impact of epigenomic profiling should be integrated in more detail. Especially in glioma, methylation profiling has been described in great detail (PMID 34594037; PMID 31023364) compared to other solid tumors.

Author Response

I greatly appreciate the helpful criticisms and suggestions from the reviewers. By including the recent studies on methylome profiling and single-cell analysis, the revision has provided a mechanistic model depicting the negative impact of CDKN2A deletion on IDH mutation-mediated inhibition of stem-like cell proliferation, thereby driving glioma progression.

Reviewer 2

“In his revised manuscript „ The Favorable Outcome of IDH-Mutant Glioma: CDKN2A/B Changes the Rule” Huang reports on elemental renewals in the revised WHO classification of tumors of the central nervous system or more specifically the upgrade of IDH-mutant astrocytoma with concurrent homozygous CDKN2A/B deletion (HD) to CNS WHO grade IV. Moreover, the author carved out further insights regarding the pathophysiological biology of how CDKN2A/B HD lead to unfavorable outcomes in this entity. I really enjoyed reviewing this manuscript which is clearly written, well-structured and of clinical relevance. In order to ensure the validity of this review article, there is one minor point that need to be addressed prior to publication. From my perspective this manuscript provides an excellent overview of fundamental modifications concerning the latest WHO-classification of gliomas.”

Response: I greatly appreciate Reviewer 2’s enthusiastic comments and the suggestion of addressing CDKN2A/B homozygous deletion in relation to epigenomic profiling (see below) to improve the quality of the manuscript.

“There exist several comprehensive methylome-analyses of IDH-mt gliomas in literature describing methylation subgroups (C-GIMP high vs. C-GIMP low; PMID 31667475 and/or PMID 26824661). In order to describe underlying molecular mechanisms of the diverging clinical behavior in IDH-mutant gliomas and the dependence on CDKN2A/B HD the impact of epigenomic profiling should be integrated in more detail. Especially in glioma, methylation profiling has been described in great detail (PMID 34594037; PMID 31023364) compared to other solid tumors.”

Response: A new paragraph (line 157 to line 175) has been added to include these suggested publications and review additional publications on G-CIMP-low glioma, CDKN2A deletion, and a developmental hierarchy from single-cell analyses of IDH-mutant glioma. Furthermore, Figure 2 has been revised to present a mechanistic model where CDKN2A deletion abrogates IDH mutation-mediated inhibition of stem-like cell proliferation.

Round 2

Reviewer 1 Report

The revision has largely been adapted to the reviewer’s suggestions.

Author Response

"The revision has largely been adapted to the reviewer’s suggestions."

Response: I appreciate Reviewer's positive comment.

This manuscript is a resubmission of an earlier submission. The following is a list of the peer review reports and author responses from that submission.

Round 1

Reviewer 1 Report

This perspective detailly discussed the exception of CDKN2A/B homogenous deletion among the favorable clinical outcome of IDH-mutant gliomas. The author highlighted the poor survival in IDH-mutant glioma patients with CDKN2A/B deletion, separating from CDKN2A/B wild-type IDH1 mutant glioma patients, indicating the importance of the classification of gliomas that needs to consider this genetic status. The author pointed out that CDKN2A/B deleted IDH1-mutant gliomas should be considered as grade 4 glioblastoma based on patient prognosis results. In addition, the author also discussed the mechanism of CDKN2A/B in cell cycle regulation and animal results of its deletion in combination with IDH1 mutation. Overall, this perspective is timingly with emphasis of defining glioma classification with addition of CDKN2A/B genetic status. The writing is very clear with also nice discussion of the implication of CDKN2A/B deletion in glioma biology. 

I only have a minor suggestion. Please see whether could add the mean survival time for each category of glioma types listed in Table 1, which could help to recognize the clinical outcome among different genetic background. 

Reviewer 2 Report

This review highlights the role of CDKN2A homozygous deletion in IDH mutant astrocytic tumors. Although this manuscript highlights the animal model with IDH1 mutation and show significance of CDKN2A HD to accelerate tumor formation, there is little updated information for readers. 

Reviewer 3 Report

This article presents a review of the two contradictory aspects of IDH1 mutants: the driver gene (oncogenic) aspect and the favorable prognostic factor aspect. The CDKN2A/B homozygous deletion is involved in the progression of IDH-mutant glioma. Although the author is trying to present CDKN2A/B mutation as a new target for drug discovery by taking into account clinical problems, it is somewhat questionable whether this can be established as a perspective paper. The following is a list of the reviewer’s comments.

  1. The working hypothesis proposed by the author is shown in Figure 1, but it is extremely abstract. It should be changed to a clearer diagram so that the molecular granulation mechanism can be understood.
  2. The author claims that p53 alteration and CDKN2A/B homozygous deletion have not received attention in IDH mutant studies (line 92), but several previous studies have already been published (PMIDs: 31832685, 31996992, 32385699, 33081848). The relevance of these previous studies to this paper should be clarified.